# Fetal Tibial Artery Doppler in Late IUGR Fetuses: A Longitudinal Study

**DOI:** 10.3390/jcm12010082

**Published:** 2022-12-22

**Authors:** Kristina Norvilaitė, Diana Ramašauskaitė, Daiva Bartkevičienė, Aleksandra Šliachtenko, Juozas Kurmanavičius

**Affiliations:** 1Center of Obstetrics and Gynecology, Institute of Clinical Medicine, Faculty of Medicine, Vilnius University, 03101 Vilnius, Lithuania; 2Institute of Clinical Medicine, Faculty of Medicine, Vilnius University, 03101 Vilnius, Lithuania

**Keywords:** late-onset IUGR, CPR, doppler examination, fetal tibial artery, pregnancy, intrauterine hypoxia

## Abstract

Introduction: Late-onset intrauterine fetal growth restriction (IUGR) is a common pregnancy complication diagnosed in 5–10% of pregnant women worldwide. Under the impact of hypoxia, the fetus develops a protective mechanism of adaptive changes occurring in the cerebral circulation (“brain-sparing effect”). Materials and methods: We conducted detailed longitudinal Doppler examinations and the monitoring of the fetal condition in 53 IUGR fetuses. Doppler measurements of the pulsatility index in the fetal tibial (TA-PI), umbilical (UA-PI), and middle cerebral arteries (MCA-PI) were performed, and the cerebral placental ratio (CPR) was determined on a weekly basis from the 33rd week to the birth. Results: The longitudinal analysis showed a significant increase in the TA-PI. The UA showed a plateau, but no increase was detected near term. The MCA-PI and CPR showed a progressive decrease in values from inclusion to delivery. Our findings indicate that the increase in the TA-PI was the first sign of the aggravating state of the fetus with the changes registered from the 35th week. The parameters of the UA-PI did not show significant changes, while the MCA and CPR became abnormal later from the 37th week. Conclusions: These observations can serve towards the development of guidelines for detecting the deteriorating signs and intervention timing in IUGR during late pregnancies.

## 1. Introduction

The intrauterine growth restriction (IUGR) of either early or late onset reflects a fetus with an estimated fetal weight (EFW) of less than the 10th percentile for its gestational age [1,2]. IUGR of early onset of <32 weeks or late onset of >32 gestational weeks is a common complication, with an estimated prevalence in approximately 5–10% of pregnancies [3,4,5]. Unfortunately, it has very low diagnostic rates because in many countries the uterus fundus measurement is still being used as the only follow-up tool. So far, there are no guidelines suggesting a routine screening in the 3rd trimester. However, detecting the late IUGR lowers the risk of stillbirth if the fetus is monitored intensively [4]. IUGR is caused by several pathological factors such as maternal, fetal, and placental. The most commonly diagnosed IUGR cases are related to placental dysfunction [6,7]. Other possible causes of IUGR can be changes in the placental form, macro- and microscopic vascular lesions, inflammation, and genetic alterations [8]. Nevertheless, there is also growing evidence of the occurrence of progressive fetal hypoxia and placental functional changes in late IUGR. It generally manifests as the main cause of stillbirth, neonatal mortality, and morbidity in the world [9,10,11]. Fetuses affected by late IUGR present with a brain-sparing effect while abnormal Doppler findings show the hemodynamic changes as an adaptation of the fetus to hypoxia [4]. Thus, the newborns are associated with a higher risk of adverse outcomes of neurodevelopment.

A broad range of Doppler studies have reported that the most frequently used UA Doppler cannot be relied on in evaluating the fetal condition in late-onset IUGR, whereas the MCA and CPR have shown a better response in detecting an abnormal or only a slightly abnormal fetal condition [12]. It is known that the parameters of the UA reflect the placental blood flow but do not reflect the fetal condition. Physiologically, UA resistance gradually decreases during pregnancy [13]. An increased resistance or pulsatility index of UA is a marker for placental insufficiency. The resistance of the MCA correlates with fetal oxygenation—the lower the resistance, the greater the risk of hypoxia, blood flow centralization, and worsening of the fetal condition [4]. The CPR serves as a more sensitive parameter to target fetal hypoxia directed to the fetal blood redistribution, rather than the MCA or UA separately in late-onset IUGR. However, too few studies about the diagnostic and prognostic value of the peripheral fetal vessels (i.e., tibial artery) are available. The increased resistance of the TA-PI in late IUGR as an evaluation of fetal oxygenation can be compromised by variable degrees of hypoxia, distress, and/or loss of full placental sufficiency. The studies on animals reveal similar patterns of adaptive mechanisms of circulatory changes in the late IUGR cases [7,9,14,15]. Although several studies [16,17] looked into the peripheral arteries of IUGR fetuses to assess the progression of hypoxia, there is still an open debate on which Doppler parameter in late IUGR cases shows a progressive or stable fetal condition.

In this study, we performed detailed longitudinal Doppler measurements of fetal arteries and analyzed the changes in Doppler parameters of each artery separately in fetuses with confirmed late IUGR. The objective of this study was the comparison of the longitudinal changes in Doppler velocity indexes of different arteries, including the peripheral fetal tibial artery to predict and monitor fetal hypoxia in late IUGR. Our findings, alongside the findings presented in similar studies, can serve as a baseline to identify key elements of monitoring late-onset IUGR and the timing of intervention that can optimize obstetrical care in forthcoming IUGR therapies for this condition.

## 2. Materials and Methods

This prospective observational study was conducted at the Center of Obstetrics and Gynecology of Vilnius University Hospital Santaros Clinics, Vilnius, Lithuania between May 2019 and May 2022. The study population included 53 pregnant women with singleton IUGR pregnancy at the gestational age from 33 + 0 to 40 + 0 weeks. IUGR was defined as an estimated fetal weight (EFW) and/or abdominal circumference (AC) <10th percentile. The gestational age was confirmed via the first-trimester ultrasound or the date of the regular menstrual period.

Over the course of the study, the data on maternal age, ethnicity, prior pregnancies, and labor outcomes, as well as the women’s medical conditions and received treatment, were collected (Table 1). The information was obtained from the internal database of the clinics. The subjects of interest included: mode of delivery (spontaneous vaginal delivery, instrumental delivery, or cesarean section), fetal amniotic fluid color, gestational age, gender, birth weight and length, 5 min-Apgar scores, umbilical cord pH values and neonatal outcomes (neonatal jaundice, neonatal hypoglycemia, need for respiratory support) (Table 2).

All the study participants underwent a weekly CTG nonstress test (NST) and a Doppler examination with the GE Healthcare Voluson E8 system. Our management protocol included the biometry and Doppler waveform pulsatility indices (PI) of the middle cerebral artery (MCA), the umbilical artery (UA), the cerebral-placental ratio (CPR), and the tibial artery (TA).

Doppler signals from the arteries under analysis were obtained as follows: the insonation angle was focused below 45 degrees in order to record the umbilical artery (UA-PI). Doppler signals were obtained from the free-floating portion of the umbilical cord. The volume of each sample was located so that it covered the artery lumen, aiming to avoid the umbilical vein. The fetal MCA was focused on the transverse view of the skull of the fetus so that the Doppler signals were obtained from the MCA which was located nearest to the transducer. The sample volume of spectral Doppler was focused on the proximal third of the MCA, near its origin site in the circle of Willis. The insonation angle was as near as possible to 0 degrees.

Furthermore, the cerebral-placental ratio (CPR) was calculated after the collection of the data. It was calculated by dividing the MCA Doppler flow by the umbilical artery UA Doppler flow.

Measurements of the TA-PI were taken using Doppler on the lower extremity where the tibial and fibular bones were clearly visible. The angle between the transducer and the bones was adjusted to 45° or less. The color Doppler gate was placed over the vessel in the leg between the two bones to locate the anterior TA (Figure 1). We were able to measure the velocity of the TA in all fetuses, signals were recorded over at least 5–6 cycles with an equal shape and amplitude of the blood flow waveforms (Figure 2 and Figure 3).

The PI of the anterior tibial artery was compared with the TA-PI standards published by Wisser et al. [17]. The estimated fetal weight was calculated using the Hadlock equation [18,19].

To explore possible abnormal neonatal outcomes, we checked for at least one of the following: Apgar score < 6 at 5 min, umbilical artery pH < 7.35, neonatal hypoglycemia or the need for respiratory support. In addition, we also evaluated the mode of delivery, the need for the induction of labor and/or the cesarean section (Table 2).

## 3. Statistical Analyses

All the analyses were performed using the SPSS software package (SPSS Inc, Chicago, IL, USA). Two-sided *p* values < 0.05 were considered statistically significant. All Doppler parameters were transformed into *Z*-values according to normative references [20,21]. The longitudinal values changes endpoint was defined as an abnormal Doppler value (MCA-PI and CPR <5th centile, UA-PI and TA-PI *>* 95th centile) [17,20,21,22,23].

The regression equations of the 5th and 95th centiles were determined by: UA PI [22], CPR [23], MCA PI [23], TA [17].

## 4. Ethics Statement

This study was approved by the Vilnius Regional Biomedical Research Ethics Committee in Vilnius on May 27, 2019 (reference No. 2019/5-1137–624).

## 5. Results

The cohort study consisted of 61 Caucasian women (the mean age at the study baseline was 32.6 ± 4.5 years). Late-onset IUGR was diagnosed in all of them. Eight subjects were excluded due to newborn weight above the 10th percentile. There were 159 scans performed on the 53 IUGR fetuses that were included and examined 3 times. The mean gestational age of subject inclusion was 35 weeks, while the mean delivery time was 37 weeks + 6 days. The last examination was performed no later than one week before delivery. There were no stillbirths or neonatal deaths in the study. The mean 5-min Apgar score was 9.49 ± 0.54.

Our results suggest that a remarkable constriction of fetal peripheral arteries can be a possible adaptation before the blood flow centralization. TA-PI changes are characterized by a progressive increase in the progressing fetal hypoxia and as a mechanism of cerebral circulation for the development of adaptive changes. First, we found that the pulsatility index in the fetal tibial (TA-PI) artery worsened significantly from the 35th week in most IUGR patients (Figure 4). Secondly, Figure 5 shows the increasing proportion of abnormal cases in the umbilical artery (UA-PI), the middle cerebral artery (MCA-PI), the cerebroplacental ratio (CPR) and the tibial artery (TA-PI). The comparison of abnormal proportions observed between the arteries in the group of 33 and 35 weeks showed no statistically significant changes in the PI of the arteries. The most significant pathological changes were found in the TA-PI of the subjects belonging to the group of 36 and 37 weeks and the group of 38 and 40 weeks. The second proportion of pathological changes was detected in the CPR and MCA-PI. The least abnormal proportion was found in the UA-PI.

## 6. Discussion

Here we provide a detailed description of the changes in the Doppler parameters of the fetal central arteries (MCA, UA), CPR, and the peripheral artery (TA) in late IUGR patients. To the best of our knowledge, this is the first longitudinal study of late IUGR fetuses with a mean follow-up period of five weeks conducted to identify, compare and analyze the changing patterns of the Doppler parameters of the central and peripheral arteries. This study also focuses on delineating the clinical features and the best timing for IUGR fetuses for optimizing the timing of the delivery.

Along the same lines, previous studies show the elevation of the tibial artery PI in IUGR fetuses from 36 weeks of gestation [17]. Moreover, as revealed by the study of 23–42 GA fetuses, monitoring of the TA-PI allows for the detection of early alterations in fetal blood circulation [16].

Our study results showed that the UA did not show significant changes near term. Previous studies reported that the parameters of the UA reflect the placental blood flow but do not reflect the fetal condition in late IUGR [24]. The absent or reversed umbilical artery end-diastolic velocity and the elevated RI correspond to the placental malperfusion and an increased risk of fetal death in early-onset IUGR cases [4,25,26]. However, ultrasound findings of UA in late-onset IUGR may show normal blood flow and cannot be a reliable indicator for fetal hypoxia [4,26,27,28]. Therefore, measurements in other fetal blood vessels are important in estimating perinatal risks. The resistance of the middle cerebral artery decreases due to fetal hypoxemia even if the resistance of UA shows normal rates [27,29,30]. Furthermore, our study reveals that the MCA-PI and the CPR showed a progressive decrease from inclusion to delivery. Our findings indicate that the MCA-PI showed fewer abnormal values than those obtained after combining it with the CPR. In particular, the Z-scores CPR and MCA were close to pathological values, but more significant changes were found in the CPR than in the MCA alone. The most significant changes were found in the TA-PI progressing with the increasing gestational age. As reported by recent ISUOG guidelines, the CPR is the parameter most recommended to monitor in late-onset IUGRs. Our study found that the use of isolated UA Doppler is likely to show a lower or no evaluation of the fetal condition, while a more advanced measurement is the CPR (MCA/UA = CPR), whereas the TA-PI Doppler measurement suggests that changes in the peripheral arteries are detected earlier. Moreover, the changes in the TA-PI were more demonstrative.

Limitations of the study. Our study focused on a small sample size of subjects, therefore no cases of stillbirth or deep hypoxia occurred. Further research into IUGR should include multi-focus research leading to appropriate conclusions and provision of recommendations for the management of late-onset IUGR and pregnancy termination due to worsening fetal well-being.

Strengths of the study. The study is unique in terms of its methodology, the subject of the study, and the focus on peripheral fetal circulation. The measurements were conducted by a single researcher using the same equipment.

## 7. Conclusions

This study reports a tendency of an accelerating PI of fetal tibial arteries in late-onset IUGR pregnancies and its association with adverse perinatal outcomes such as neonatal acidemia and the need for respiratory support. Our findings sustain that the alterations in the TA-PI are detected earlier in comparison with other traditional Doppler parameters like the MCA-PI, the CPR, and the UA-PI, and can be the sign of fetal blood centralization and hypoxia. Ultimately, TA-PI Doppler measurements can be used as an additional diagnostic tool together with methods already used for an early prediction of fetal distress in late IUGR.

## Figures and Tables

**Figure 1 jcm-12-00082-f001:**
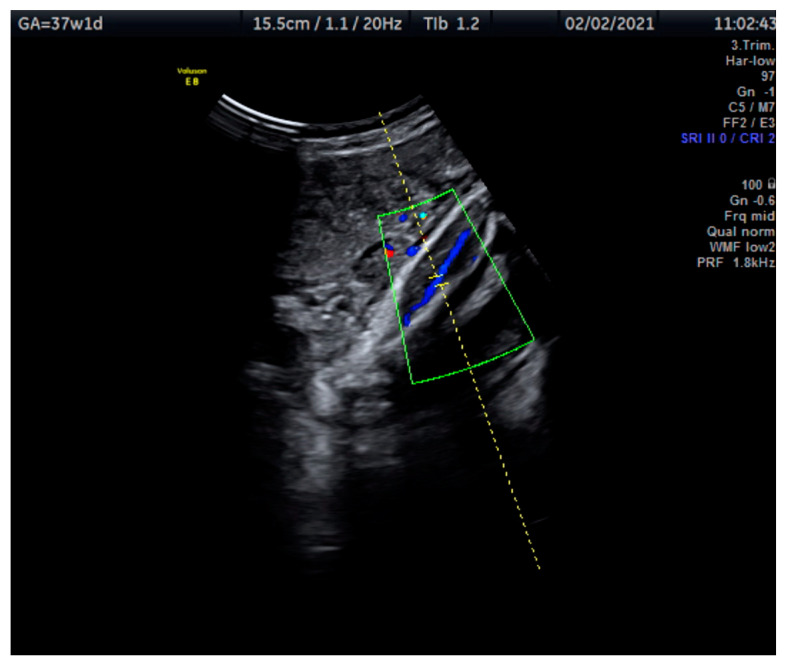
This Doppler image shows the technique of the measurement of the blood flow of the tibial artery of the fetal lower extremity with the tibial and fibular bones clearly visible. The measurement was taken at 37 + 1 weeks.

**Figure 2 jcm-12-00082-f002:**
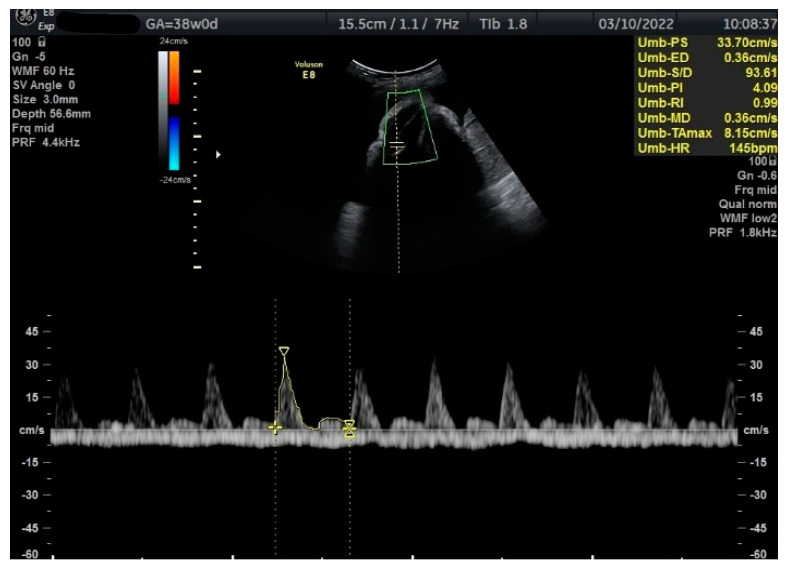
Doppler measurement at 38 + 0 weeks of normal TA-PI (measured manually).

**Figure 3 jcm-12-00082-f003:**
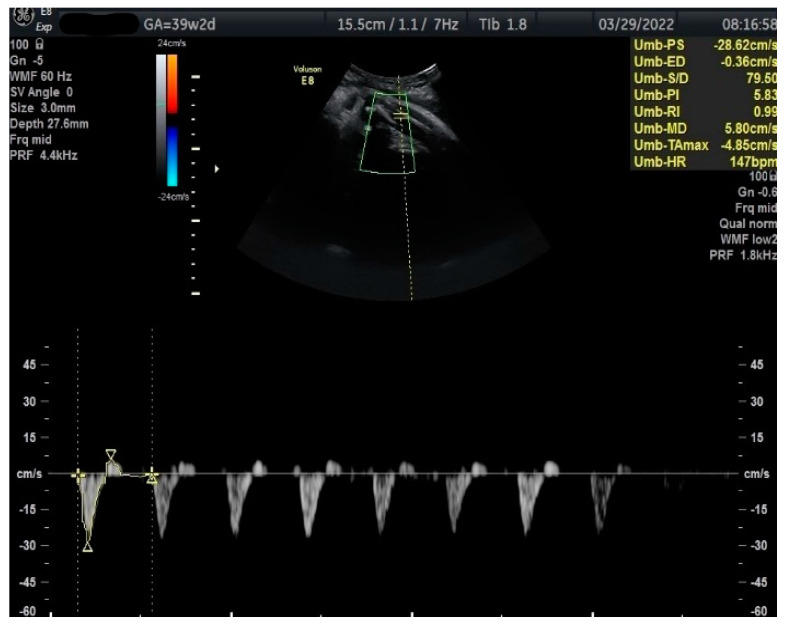
Doppler measurement at 39 + 2 weeks of TA-PI shows reverse flow >95th percentile (measured manually).

**Figure 4 jcm-12-00082-f004:**
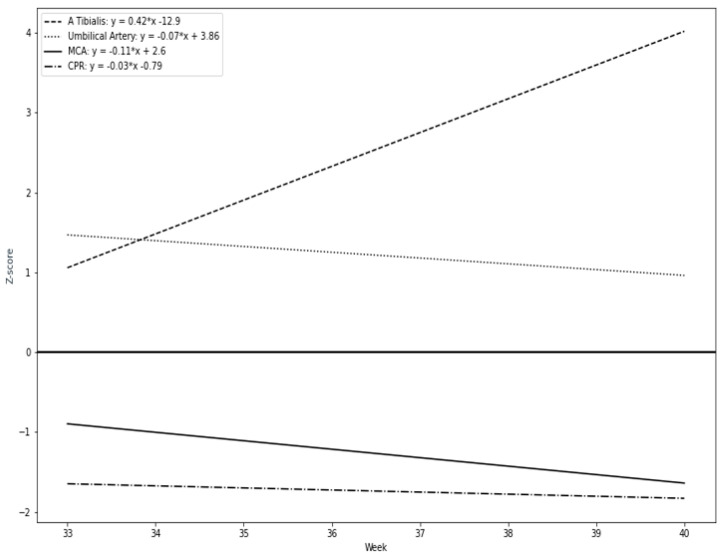
Longitudinal trends of Doppler parameters in Z-values during the study period: tibial artery (TA-PI), cerebroplacental ratio (CPR), middle cerebral artery (MCA-PI), umbilical artery (UA-PI).

**Figure 5 jcm-12-00082-f005:**
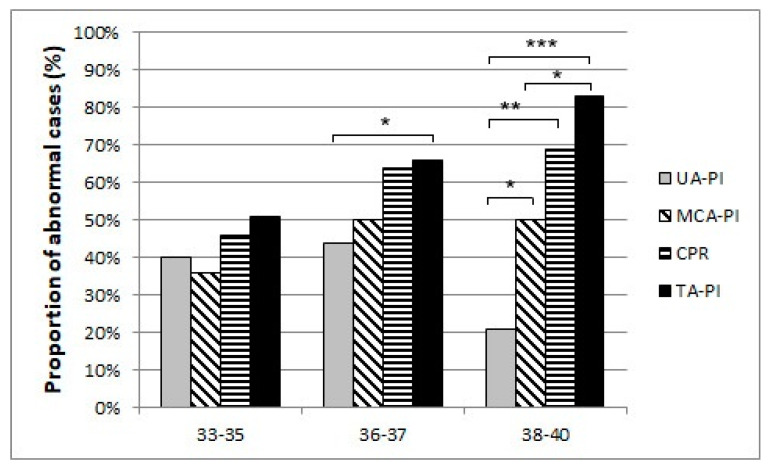
The proportion of abnormal cases in the umbilical artery (UA-PI), the middle cerebral artery (MCA-PI), cerebroplacental ratio (CPR), and the tibial artery (TA-PI) Doppler findings between 33 and 35, 36 and 37, 38 weeks of gestation and last examination before delivery (*p*-Values *** *p* < 0.001, ** *p* < 0.01, * *p* < 0.05).

**Table 1 jcm-12-00082-t001:** Subject characteristics in pregnant women with IUGR.

Characteristics	Participants (Pregnant Women with IUGR)
Maternal age	32.6 ± 4.5 (years)
Ethnicity (Caucasian)	53 (100%)
Parity	
Nulliparous	27 (51%)
Multiparous	26 (49%)
Comorbidity	
- Preeclampsia	9 (17%)
- Gestational hypertension	6 (11%)
- Pregestational hypertension	2 (4%)
Mode of delivery	
- CS	15 (28%)
- Natural	23 (42%)
- Induced labor	13 (25%)

**Table 2 jcm-12-00082-t002:** Perinatal outcomes of IUGR newborns.

Characteristics	IUGR Newborn
Gestational age at birth	37.6 ± 1.6
Neonate gender	
- Male	23 (43%)
- Female	30 (57%)
Birth weight	2369.6 ± 409
Birth weight < 5 percentile	33 (62.3%)
Meconium-stained amniotic fluid	2 (3.8%)
Meconium aspiration syndrome	0
Umbilical artery pH	7.31 ± 1.02
Umbilical artery pH < 7.35	36 (68%)
Umbilical artery pH < 7.2	5.0 (9.4%)
Umbilical artery pH < 7	0
Apgar score at 5 min	9.49 ± 0.54
Apgar at 5 min ≤ 3	0
Apgar at 5 min < 6	0
Stillbirth	0
Need for respiratory support at birth	5 (9.4%)
Neonatal jaundice	14 (26.4%)
Neonatal hypoglycemia	6 (11.3%)

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
