# Peer review of "Fetal Tibial Artery Doppler in Late IUGR Fetuses: A Longitudinal Study"

_jcm, 2022, doi:10.3390/jcm12010082_

Round 1

Reviewer 1 Report

This study by Norvilaite et al. entitled "Fetal Tibial Artery Doppler in Late IUGR Fetuses: A Longitudinal Study" describes the use of Doppler sonographic parameters for late growth retardation in fetuses. In doing so, the authors show longitudinal data for various Doppler-parametric vessels that they observe over time. The tibial artery is also examined and evidence is found that this vessel is the first to be affected in late IUGR. The language style of the manuscript is good and the data well presented. However, in the ultrasound image the angle for the Doppler is not correct, so I would like to see a more accurate presentation of the Doppler parameters here. Also what settings were used for the titbial artery ? Unfortunately, the authors do not describe the outcome of the fetuses, which would be desirable in the manuscript. The authors should also describe the devices, presets and settings used in the methodology in more detail. Some descriptions are missing in the methodology, so that it is not complete. Therefore, the manuscript would have to be revised before it can be released for publication. 

Author Response

Dear Reviewer,

We are grateful for your time and effort in delivering an in-depth review of our paper. Please find my responses to your comments below.

Comments from Reviewer 1 

Comment 1: However, in the ultrasound image the angle for the Doppler is not correct, so I would like to see a more accurate presentation of the Doppler parameters here. Also what settings were used for the tibial artery? 

Response:  We agree with your suggestions and have supplemented our manuscript with the following information. When measuring the pulsatility index (PI), the angle for the Doppler does not need to be adapted because the angle has no influence on PI value. However, when measuring the blood flow velocity in the vessels, the angle is important. In our study, the parameter was 0 because we were measuring only PI and we were not adjusting the insonation angle. Considering the fact that this model of the ultrasound system is not equipped with the software for measuring the tibial artery, we conducted all the measurements of the tibial artery with the settings fixed for the umbilical artery.

Please see the reference below:

https://onlinelibrary.wiley.com/doi/abs/10.7863/jum.2006.25.9.1187

Comment 2: Unfortunately, the authors do not describe the outcome of the fetuses, which would be desirable in the manuscript. 

Response: Perinatal outcomes are presented in Table 2, page 3 of the manuscript. We also added more details on newborns in providing the perinatal outcomes. However, the analysis of perinatal outcomes was out of the scope of this article and is planned to be presented in the forthcoming paper. 

Comment 3: The authors should also describe the devices, presets and settings used in the methodology in more detail. Some descriptions are missing in the methodology, so that it is not complete.

Response: Thank you for your insight and this important suggestion. We agree with your suggestions and have supplemented our manuscript with the following information. Doppler examination of all the subjects was conducted with the Voluson E8 ultrasound system. The tibial artery parameters are described in detail in the methodology section of our manuscript using methods referred to in literature source No. 17, page 4 of the manuscript. The method used to measure the umbilical artery (UA) was as follows: Doppler ultrasonography was performed with the ultrasound system transducer Voluson E8. Color Doppler imaging was used to optimize the insonation obtained by the pulsed Doppler examination. The angle of insonation was kept at less than <45 degrees in all cases. The Doppler velocity waveforms were obtained from the free-floating loop of the umbilical cord during quiescence. Five to 6 uniform waveforms were obtained >= 3 times in succession, and automatically measurements were calculated. For more information, please refer to source No. 22.

The MCA-PI was measured using the techniques provided by literature source No. 23. Doppler ultrasound measurements were recorded using the transabdominal transducer with Voluson E8 ultrasound system. The MCA was visualized using color flow mapping in an axial section of the brain. The Doppler beam was directed along the MCA, and the sample volume and was placed over the proximal section where the MCA emerges from the circle of Willis. The recordings were acquired in the absence of fetal breathing or body movements over at least three uniform heart cycles. The Doppler waveforms were traced and PI calculated automatically. The cerebroplacental ratio was calculated as the ratio of the MCA-PI and UA-PI.

 Additional clarifications

In addition to the above comments, the use of the language has been additionally revised by the English language professional. 

We look forward to hearing from you in due time if any further questions or comments you may have. 

Sincerely,

Kristina Norvilaite

12/14/2022

Reviewer 2 Report

The paper is scientifically interesting and original, it is presented in a well-structure manner, bringing up a new parameter that could help diagnose late IUGR. Larger studies should be conducted to better understand the importance of the fetal tibial artery Doppler in late IUGR.

Abstract: The abstract is informative and properly structured.

Keywords: The keywords are significant to the subject.

Introduction: It is necessary to define the late-onset IUGR and early-onset IUGR.

Materials and methods are well presented and described, offering the necessary details to reproduce the study.

Results: the results are properly described, the figures and tables are appropriate, properly showing the data and are easy to interpret.

Discussion: The comments in this section are focused of the subject with adequate and up-to-date references.

References: The references are mostly recent publications and relevant

Author Response

Dear Reviewer,

We are grateful for your time and effort in delivering an in-depth review of our paper. Please find my responses to your comments below.

Comments from Reviewer 2 

Comment 1:

The paper is scientifically interesting and original, it is presented in a well-structure manner, bringing up a new parameter that could help diagnose late IUGR. Larger studies should be conducted to better understand the importance of the fetal tibial artery Doppler in late IUGR.

Response: Thank you for your favorable review. We appreciate your accurate evaluation of our study and agree with the recommendations on the need to further investigate the fetal tibial artery Doppler parameters in late IUGR so that more clinical evidence in support of a more relevant diagnosis is obtained. 

Comment 2: Introduction: It is necessary to define the late-onset IUGR and early-onset IUGR.

Response: Thank you for pointing this out. We updated the introduction section with the gestational age range for both, early and late onset IUGR, page 1, in the Introduction section. 

 Additional clarifications

In addition to the above comments, the use of the language has been additionally revised by the English language professional. 

We look forward to hearing from you in due time if any further questions or comments you may have. 

Sincerely,

Kristina Norvilaite

12/14/2022